# Advancing Radiograph Representation Learning with Masked Record Modeling

**Hong-Yu Zhou**[1,2*]  **Chenyu Lian**[1*]  **Liansheng Wang**[1†]  **Yizhou Yu**[2,3†]

[1]School of Informatics, Xiamen University
[2]Department of Computer Science, The University of Hong Kong
[3]AI Lab, Deepwise Healthcare
`whuzhouhongyu@gmail.com, cylian@stu.xmu.edu.cn,`
`lswang@xmu.edu.cn, yizhouy@acm.org`

## Abstract

Modern studies in radiograph representation learning ($R^2L$) rely on either self-supervision to encode invariant semantics or associated radiology reports to incorporate medical expertise, while the complementarity between them is barely noticed. To explore this, we formulate the self- and report-completion as two complementary objectives and present a unified framework based on masked record modeling (MRM). In practice, MRM reconstructs masked image patches and masked report tokens following a multi-task scheme to learn knowledge-enhanced semantic representations. With MRM pre-training, we obtain pre-trained models that can be well transferred to various radiography tasks. Specifically, we find that MRM offers superior performance in label-efficient fine-tuning. For instance, MRM achieves 88.5% mean AUC on CheXpert using 1% labeled data, outperforming previous $R^2L$ methods with 100% labels. On NIH ChestX-ray, MRM outperforms the best performing counterpart by about 3% under small labeling ratios. Besides, MRM surpasses self- and report-supervised pre-training in identifying the pneumonia type and the pneumothorax area, sometimes by large margins. Code and models are available at `https://github.com/RL4M/MRM-pytorch`.

## 1 Introduction

Radiograph representation learning ($R^2L$) has been among the core problems of medical image analysis. Previously, downstream radiograph analysis tasks counts on pre-trained models on ImageNet (Deng et al., 2009) or large X-ray datasets (Wang et al., 2017; Irvin et al., 2019; Johnson et al., 2019; Bustos et al., 2020) to alleviate the shortage of expert labeling. The emergence of self-supervised representation learning (Doersch et al., 2015; Agrawal et al., 2015; Wang & Gupta, 2015; Zhou et al., 2021a; 2023) provides a choice to conduct pre-training with negligible human intervention by exploiting self-supervision. However, the self-supervised paradigm ignores the introduction of medical expertise (e.g., anatomy), reducing its transferability to downstream tasks with limited label information.

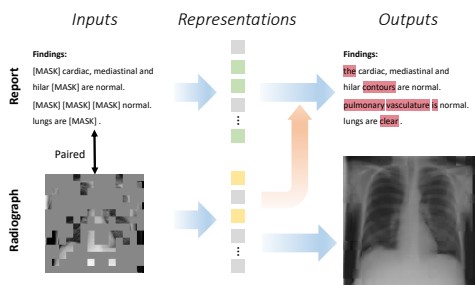

Figure 1: Illustration. MRM learns transferable radiograph representations via reconstructing masked records, i.e., masked radiograph patches and masked reports tokens.

On the other hand, free-text radiology reports written by experienced radiologists often contain rich domain knowledge. To leverage this, researchers developed automated rule-based labelers (Wang

*Work done while visiting Xiamen University. First two authors contributed equally.
†Corresponding authors.

et al., 2017; Irvin et al., 2019) to extract structured labels from unstructured texts. Nevertheless, these labelers have several practical limitations. First, some procedures of the label extraction workflow, such as rulemaking and natural language processing, still require the intensive involvement of experts and engineers. Besides, the developed labelers can hardly adapt to new scenarios due to the fixed rules and lexicons.

Against this background, report-supervised $R^2L$ was proposed (Zhang et al., 2020) to acquire supervision from radiology reports. In practice, this paradigm leverages words and sentences in free-text reports as supervision to guide deep neural networks to learn radiograph representations, outperforming the archetypical label- and self-supervised pre-training by observable margins in various downstream tasks (Zhang et al., 2020; Zhou et al., 2022). The report-supervised $R^2L$ highlights the importance of the incorporation of domain knowledge. This differs from the self-supervised paradigm, which focuses on learning invariant semantic representations. Nonetheless, current studies view the self- and report-supervised $R^2L$ as separate, discrete choices, preventing their combinations.

Driven by this analysis, we present a unified framework based on masked record modeling (MRM), where the self- and report-completion tasks are modeled as two complementary objectives. Specifically, masked image reconstruction integrates semantics into pre-trained models, while masked report restoration facilitates the incorporation of medical expertise. As a result, MRM learns knowledge-enhanced semantic representations that generalize well. In practice, MRM masks random patches and tokens from the input radiograph and associated radiology report with high masking ratios. Following a multi-task scheme, MRM asks the radiography pre-trained model to learn visual representations that can not only reconstruct the missing patches but also restore the missing tokens from the non-masked token embeddings along with mask tokens.

With MRM pre-training, we can train radiography models on MIMIC-CXR (Johnson et al., 2019) with improved generalization performance. With a pre-trained ViT-B/16 model, we achieve 88.5% mean AUC when fine-tuned on CheXpert (Irvin et al., 2019) with only 1% labels. This outperforms all previous counterparts with 100% labeled data. On NIH ChestX-ray (Wang & Gupta, 2015), MRM surpasses the report-supervised paradigm by about 3% when the labeling ratios[1] are 1% and 10%. On pneumonia identification tasks, MRM outperforms self- and report-supervised baselines, sometimes by substantial margins. These observations help verify the effectiveness of MRM in learning more transferable radiograph representations.

## 2 RELATED WORK

### 2.1 REPORT-SUPERVISED RADIOGRAPH REPRESENTATION LEARNING

Recently, report-supervised learning (Zhang et al., 2020; Liao et al., 2021; Huang et al., 2021; Zhou et al., 2022; Boecking et al., 2022) emerges as a new $R^2L$ paradigm that automatically acquires supervision from free-text radiology reports. Zhang et al. (2020) proposed ConVIRT to contrast the radiograph features with latent embeddings of sentences in radiology reports. Liao et al. (2021) and Huang et al. (2021) explored the alignment between local patches and words in the report. Zhou et al. (2022) presented a Transformer-based $R^2L$ framework that conducts autoregressive report modeling and study-report matching. Report-supervised $R^2L$ takes the advantage of label-supervised learning, which is the incorporation of domain knowledge. Compared to the self-supervised paradigm, report-supervised $R^2L$ lays no emphasis on learning semantically invariant representations. To address the discrepancy between them, we formalize self- and report-completion as two complementary objectives, based on which we propose to encode both semantics and medical expertise into latent representations following a multi-task scheme.

### 2.2 VISUAL REPRESENTATION LEARNING VIA IMAGE-LANGUAGE PRE-TRAINING

Learning visual representations from image-language pairs has achieved tremendous success in natural image tasks (Sariyildiz et al., 2020; Desai & Johnson, 2021; Radford et al., 2021; Mu et al., 2021; Zhao et al., 2022; Li et al., 2021; Geng et al., 2022; Wang et al., 2022; Chen et al., 2022;

---

[1] The labeling ratio X% means that X% of the training set from a fully annotated downstream dataset are used for supervised fine-tuning.

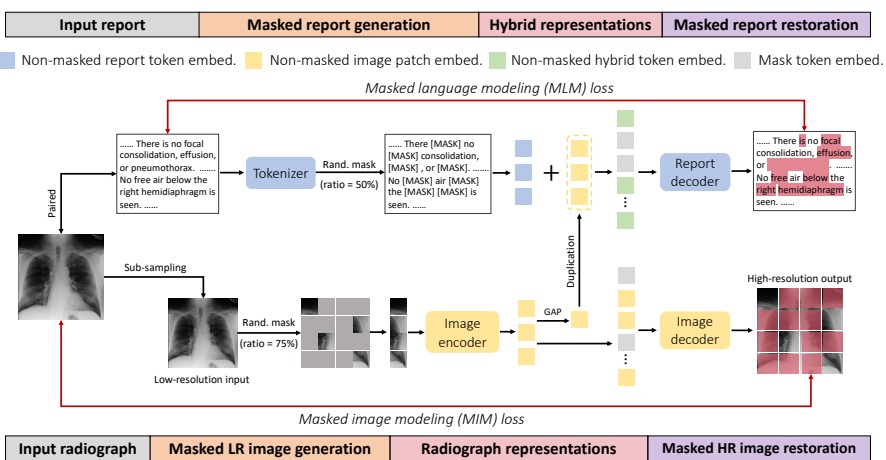

Figure 2: OVERVIEW. During the pre-training stage, MRM requires the image encoder to provide radiograph representations to simultaneously support the restoration of masked radiograph patches and masked associated radiology report tokens. The masked language and image modeling losses are only calculated on image and report tokens highlighted in pink. **Embed.**, **LR**, and **HR** stand for embeddings, low-resolution, and high-resolution, respectively.

Singh et al., 2022; Yu et al., 2022; Dou et al., 2022; Arici et al., 2021; Kwon et al., 2022). Similar to ConVIRT (Zhang et al., 2020), image-language contrast has been widely adopted to conduct pre-training (Radford et al., 2021; Mu et al., 2021; Li et al., 2021; Yu et al., 2022; Dou et al., 2022; Gao et al., 2022). Nowadays, efforts have been made to train a unified encoder for vision and language data (Geng et al., 2022; Wang et al., 2022; Chen et al., 2022; Geng et al., 2022). Akin to our approach, SLIP (Mu et al., 2021) combines SimCLR (Chen et al., 2020) and CLIP (Radford et al., 2021) to train a vision encoder using image-language pairs. However, SLIP only slightly outperforms SimCLR in fine-tuning, while requiring large batch sizes and tens of millions of image-language pairs for pre-training. In contrast, our MRM surpasses various self-supervised methodologies by large margins and can be pre-trained using only hundreds of thousands of radiograph-report pairs, enabling effective medical visual representation learning with limited annotations and computing resources.

## 3 MASKED RECORD MODELING

We propose MRM (i.e., Masked Record Modeling) to learn radiograph representations using record-level supervision. As the name implies, MRM acquires supervision signals from both radiographs and associated radiology reports. The motivation behind is to learn knowledge-enhanced semantic latent representations by reconstructing masked radiograph patches and masked radiology report tokens in medical records.

Fig. 2 presents an overview of MRM. We first apply random masking to each low-resolution radiograph and its associated radiology report (with different high masking ratios). Then, we forward the obtained non-masked image patches to the image encoder to acquire non-masked image patch embeddings. These embeddings serve two purposes: (**i**) assist non-masked report tokens to restore the masked report tokens; (**ii**) restore the high-resolution masked radiograph patches. To achieve the first goal, we add the globally averaged radiograph representation to each non-masked report token embedding and pass the resulting hybrid representations to the report decoder for masked report restoration. As for the second purpose, we conduct a novel patch restoration task to explicitly encode more local details into radiograph representations by reconstructing high-resolution patches from low-resolution inputs.

## 3.1 REPORT COMPREHENSION

**Masked report generation.** In our scenario, each radiology report is associated with a radiograph. To convert the free-text report into tokens, we use WordPiece (Wu et al., 2016) as the default tokenizer, whose vocabulary has approximately 30k tokens. After tokenization, we randomly mask a number of report tokens with [MASK]. Compared to BERT (Devlin et al., 2018) that randomly masks 15% tokens, we use a 50% probability of masking each token in the report. The insight behind the use of a higher masking ratio is that we want the model to lean more upon the image embeddings to finish the report-completion task.

**Hybrid representations for storing multi-modal information.** We then transform non-masked report tokens into token embeddings using a simple lookup table[2], which stores randomly initialized embeddings of a fixed dictionary and size. In practice, we retrieve embeddings using indices. Then, the global embedding of the associated radiograph is added to each non-masked token embedding. The resulting non-masked hybrid embeddings are supposed to include the multi-modal information from the radiograph and associated radiology report, which ought to be helpful for restoring the masked tokens.

**Masked report restoration.** To reconstruct the masked tokens, we forward latent embeddings of both hybrid tokens and mask tokens to the report decoder (a light-weight transformer model), where fixed (i.e., unlearnable) positional embeddings are added to encode the position information. We train the report decoder using the masked language modeling objective.

## 3.2 RADIOGRAPH UNDERSTANDING

**Masked image generation with low resolution.** We propose to learn radiograph representations by reconstructing high-resolution radiograph patches from low-resolution inputs. The motivation behind is to encode more local information into latent embeddings via super-resolution imaging. As shown in Fig. 2, we sub-sample each high-resolution radiograph by a factor of two to generate a low-resolution input. Following He et al. (2022), we split low-resolution radiograph into non-overlapping image patches, where 75% patches are randomly masked.

**Radiograph representations.** We add fixed unlearnable positional embeddings to linearly transformed non-masked image patches. Next, we forward the resulting patch embeddings to the transformer-based image encoder, which produces non-masked image patch embeddings. Then, the global average pooling (GAP) is applied to all non-masked embeddings, whereby a global feature is obtained. Here, we hypothesize that the image-level information brought by the global feature is helpful to the restoration of masked report tokens. Based on this hypothesis, we duplicate and add the global feature to each non-masked report token embedding, producing the hybrid token embeddings that encode the multi-modal information.

**Masked image restoration with high resolution.** Non-masked image and mask token representations with added positional embeddings are passed to the image decoder for the restoration of masked radiograph patches. Specifically, the image decoder is required to restore a high-resolution ($2\times$ the input resolution) patch from each input token via a shared fully-connected (FC) layer (across all tokens). In practice, the proposed restoration procedure explicitly requires the learned image representations to include more local details that often matter a lot in medical diagnosis.

## 3.3 MULTI-TASK MODELING

Suppose each input radiograph consists of two set $\mathcal{I}_{\mathcal{M}}$ and $\mathcal{I}_{\mathcal{N}}$. The masked set $\mathcal{I}_{\mathcal{M}}=\{\mathbf{x}_1,\ldots,\mathbf{x}_h\}$ (ground truths) contains $h$ high-resolution image patches that serve as reconstruction targets. The non-masked set $\mathcal{I}_{\mathcal{N}}=\{\mathbf{s}_1,\ldots,\mathbf{s}_k\}$ comprises $k$ low-resolution patches that are treated as model inputs. Likewise, we denote the associated radiology report using the masked set $\mathcal{R}_{\mathcal{M}}=\{\mathbf{u}_1,\ldots,\mathbf{u}_p\}$ (ground truths) and the non-masked set $\mathcal{R}_{\mathcal{N}}=\{\mathbf{v}_1,\ldots,\mathbf{v}_q\}$ (inputs). Here, $\mathbf{x}$, $\mathbf{s}$, $\mathbf{u}$, and $\mathbf{v}$ stand for the masked image patch, non-masked image patch, masked report token, and non-masked report token, respectively. For model parameters, we use $\Theta_E$, $\Theta_D$, and $\Theta_R$ to denote the parameters of the image encoder, image decoder, and report decoder, respectively.

---

[2]https://pytorch.org/docs/stable/generated/torch.nn.Embedding.html.

For the restoration of masked report tokens, we forward hybrid representations to the report decoder and minimize the negative log-likelihood function. During the training stage, the objective function $\mathcal{L}_{\text{R}}$ (i.e., the MLM loss in Fig. 2) of the above optimization procedure can be summarized as follows:

$$\mathcal{L}_{\text{R}}(\mathcal{R}_{\mathcal{M}}, \mathcal{R}_{\mathcal{N}}, \mathcal{I}_{\mathcal{N}}) = -\sum_{i=1}^{p} \log P\left(\mathbf{u}_i \mid \mathbf{v}_{1:q}, \mathbf{s}_{1:k};\ \Theta_E, \Theta_R\right), \tag{1}$$

where $P$ stands for the conditional probability. We ignore the mask tokens for simplicity.

Similarly, we can formalize the objective function of the high-resolution masked radiograph restoration (cf. the MIM loss in Fig. 2) as follows:

$$\mathcal{L}_{\text{I}}(\mathcal{I}_{\mathcal{M}}, \mathcal{I}_{\mathcal{N}}) = \text{MSE}\left(f_{\Theta_D}(f_{\Theta_E}(\mathbf{s}_{1:k})), \mathbf{x}_{1:h}\right). \tag{2}$$

In practice, we adopt the mean squared error (MSE) to measure the differences between the predicted and ground-truth image patches with high resolution, where all pixel values are normalized to [0, 1].

The total multi-task training objective function (to be minimized) of the multi-modal restoration is as follows:

$$\mathcal{L}(\mathcal{R}_{\mathcal{M}}, \mathcal{R}_{\mathcal{N}}, \mathcal{I}_{\mathcal{M}}, \mathcal{I}_{\mathcal{N}}) = \mathcal{L}_{\text{R}}(\mathcal{R}_{\mathcal{M}}, \mathcal{R}_{\mathcal{N}}, \mathcal{I}_{\mathcal{N}}) + \lambda \mathcal{L}_{\text{I}}(\mathcal{I}_{\mathcal{M}}, \mathcal{I}_{\mathcal{N}}) \tag{3}$$

where $\lambda$ is a hyper-parameter that controls the relative impacts of two objective functions. After pre-training, we can transfer the weight parameters of the image encoder (i.e., $\Theta_E$) to various downstream tasks for fine-tuning.

## 4 EXPERIMENTS

In this section, we mainly compare MRM against report- and self-supervised $R^2L$ methodologies on 5 well-established public datasets. Average results are reported over three training runs.

### 4.1 MIMIC-CXR FOR PRE-TRAINING

We conduct pre-training on MIMIC-CXR (Johnson et al., 2019), one of the largest X-ray datasets, that contains more than 370,000 radiograph images from over 220,000 patient studies. Each radiograph is paired with one associated radiology report.

### 4.2 DATASETS FOR FINE-TUNING

We validate the transferability of learned radiograph representations on X-ray based classification and segmentation tasks via end-to-end fine-tuning. Specifically, we evaluate the pre-trained model on 4 X-ray datasets in the classification tasks, which are NIH ChestX-ray (Wang et al., 2017), CheXpert (Irvin et al., 2019), RSNA Pneumonia (Shih et al., 2019), and COVID-19 Image Data Collection (Cohen et al., 2020). For the segmentation task, we fine-tune the pre-trained model on SIIM-ACR Pneumothorax Segmentation.[3]

**CheXpert** introduces a multi-label classification problem on chest X-rays. We follow the official guideline (Irvin et al., 2019) and report the model performance on 5 selected pathologies, i.e., atelectasis, cardiomegaly, consolidation, edema, and pleural effusion. Considering the official test set of CheXpert is not available to the public, we follow ConVIRT (Zhang et al., 2020) to regard the official validation set as the test set. Meanwhile, we randomly sample 5,000 images from the official training set to build the validation set. The training/validation/test split each constitutes 218,414/5,000/234 images of the whole dataset.

**RSNA Pneumonia** defines a binary classification problem, where each chest radiograph is categorized as either pneumonia or normal. We adopt the official data split, where the training/validation/test set comprises 25,184/1,500/3,000 images, respectively.

**NIH ChestX-ray** consists of about 112,120 frontal-view chest radiograph images, where a multi-label classification problem on 14 chest pathologies is introduced. The training/validation/test split each constitutes 70%/10%/20% of the whole dataset.

---

[3]https://www.kaggle.com/c/siim-acr-pneumothorax-segmentation.

| Methods | Input Size | Pre-train. Data | CheXpert | | | RSNA Pneumonia | | | SIIM | |
|---|---|---|---|---|---|---|---|---|---|---|
| | | | 1% | 10% | 100% | 1% | 10% | 100% | 10% | 100% |
| Our MRM | 224 | MI-CXR | 88.5±0.7 | 88.5±0.6 | 88.7±0.3 | **91.3±0.6** | **92.7±0.4** | **93.3±0.4** | **73.2±0.5** | **91.4±0.3** |
| *CNN-based* | | | | | | | | | | |
| ConVIRT | 224 | CheXpert | 85.9 | 86.8 | 87.3 | 77.4 | 80.1 | 81.3 | 43.2 | 59.9 |
| GLoRIA | 224 | CheXpert | 86.6 | 87.8 | 88.1 | 86.1 | 88.0 | 88.6 | 46.9 | 63.4 |
| ConVIRT | 224 | MI-CXR | 87.0 | 88.1 | 88.1 | 88.8 | 91.5 | 92.7 | - | - |
| MedKLIP† | 224 | MI-CXR | - | - | - | 87.3 | 88.0 | 89.3 | 72.1 | 79.4 |
| BioViL | 480 | PubMed + MI-III/CXR | - | - | - | 88.1 | 88.4 | 89.1 | - | - |
| *Transformer-based* | | | | | | | | | | |
| GLoRIA* | 224 | MI-CXR | 86.5±0.8 | 87.5±0.6 | 87.8±0.5 | 89.7±0.8 | 91.2±0.5 | 92.1±0.3 | 71.8±0.7 | 90.9±0.4 |
| REFERS | 224 | MI-CXR | 87.2±0.8 | 88.1±0.5 | 88.2±0.3 | 89.4±0.7 | 91.6±0.7 | 92.7±0.4 | 72.1±0.5 | 89.7±0.2 |
| M3AE | 224 | MI-CXR | 86.2±0.6 | 87.3±0.6 | 87.9±0.4 | 89.0±0.5 | 90.8±0.6 | 92.3±0.3 | 72.0±0.7 | 90.4±0.3 |
| MGCA† | 224 | MI-CXR | 88.8 | 89.1 | 89.7 | 89.1 | 89.9 | 90.8 | 59.3 | 64.2 |

Table 1: COMPARISONS ON CHEXPERT, RSNA PNEUMONIA, AND SIIM. We report AUC scores of different labeling ratios when fine-tuning on CheXpert and RSNA Pneumonia. In comparison, dice scores are presented on SIIM. The best results are bolded. **MI-** stands for the MIMIC dataset series. Note that ResNet-50 and ViT-B/16 are treated as the default backbones for CNN- and Transformer-based methods, respectively. * denotes our implementation of GLoRIA using ViT-B/16. Approaches with † leverage disease-level annotations for pre-training. Specifically, numbers of MGCA on CheXpert and RSNA Pneumonia are linear classification results.

| Labeling Ratios | Methods | Average | Atelectasis | Cardiomegaly | Consolidation | Edema | Effusion | Emphysema | Fibrosis | Hernia | Infiltration | Mass | Nodule | Pleural Thickening | Pneumonia | Pneumothorax |
|---|---|---|---|---|---|---|---|---|---|---|---|---|---|---|---|---|
| 1% | Our MRM | **79.4±0.8** | **78.8** | **90.3** | **80.0** | **86.5** | **86.9** | **82.0** | 71.9 | **90.0** | 67.2 | **82.3** | **69.6** | **72.3** | **69.6** | **84.0** |
| | MedKLIP | 77.2 | - | - | - | - | - | - | - | - | - | - | - | - | - | - |
| | REFERS | 76.7 | 77.5 | 85.6 | 78.6 | 84.9 | 85.4 | 79.5 | **72.3** | 77.1 | **67.5** | 76.2 | 66.5 | 71.6 | 69.3 | 81.7 |
| | Model Genesis | 70.3 | 72.1 | 67.1 | 75.8 | 76.1 | 80.6 | 72.6 | 64.8 | 73.5 | 65.7 | 65.2 | 62.2 | 67.6 | 64.8 | 76.2 |
| | C2L | 71.1 | 75.1 | 67.1 | 77.6 | 75.1 | 83.4 | 71.5 | 66.8 | 70.0 | 63.8 | 70.1 | 66.2 | 68.1 | 65.7 | 74.4 |
| | Context Restoration | 67.8 | 69.1 | 64.4 | 73.2 | 73.8 | 78.1 | 70.0 | 62.1 | 70.2 | 65.2 | 62.4 | 59.1 | 65.0 | 62.2 | 73.8 |
| | TransVW | 71.3 | 74.5 | 68.9 | 76.7 | 79.8 | 81.1 | 67.9 | 68.7 | 68.2 | 66.8 | 66.5 | 66.2 | 68.5 | 68.8 | 75.0 |
| | ImageNet Pre-training | 69.8 | 73.3 | 69.6 | 76.0 | 81.7 | 80.5 | 67.1 | 64.9 | 64.8 | 65.8 | 67.0 | 62.3 | 65.7 | 65.0 | 74.0 |
| 10% | Our MRM | **84.0±0.5** | **82.3** | **90.9** | **81.1** | **89.0** | **88.8** | **92.2** | **84.8** | **94.0** | **70.1** | **86.6** | **75.1** | **78.6** | **74.3** | **88.4** |
| | MedKLIP | 78.9 | - | - | - | - | - | - | - | - | - | - | - | - | - | - |
| | REFERS | 80.9 | 80.1 | 89.8 | 79.5 | 87.8 | 87.5 | 88.2 | 77.2 | 86.1 | 69.6 | 82.0 | 72.8 | 74.2 | 72.2 | 85.6 |
| | Model Genesis | 76.0 | 77.2 | 72.8 | 77.5 | 85.7 | 85.2 | 81.0 | 75.3 | 78.0 | 68.4 | 73.1 | 69.5 | 72.2 | 67.7 | 80.4 |
| | C2L | 76.6 | 78.0 | 75.5 | 77.5 | 84.1 | 85.7 | 81.2 | 73.7 | 79.5 | 67.4 | 77.5 | 71.7 | 72.0 | 67.3 | 81.9 |
| | Context Restoration | 73.8 | 75.5 | 70.6 | 77.1 | 84.5 | 84.2 | 79.4 | 73.1 | 67.5 | 68.1 | 70.9 | 66.9 | 71.7 | 65.2 | 79.1 |
| | TransVW | 74.4 | 76.5 | 70.8 | 77.6 | 83.0 | 84.8 | 79.7 | 69.9 | 74.7 | 68.5 | 72.1 | 68.3 | 72.4 | 63.2 | 79.6 |
| | ImageNet Pre-training | 74.4 | 74.2 | 79.8 | 75.9 | 85.7 | 83.2 | 80.4 | 72.1 | 74.0 | 64.1 | 71.7 | 65.6 | 69.6 | 66.2 | 79.7 |
| 100% | Our MRM | **85.9±0.3** | **84.2** | **93.0** | **82.2** | **91.0** | **89.6** | **94.3** | **86.7** | **94.4** | 71.8 | **88.2** | **78.5** | **81.4** | **77.3** | **90.2** |
| | MedKLIP | 83.2 | - | - | - | - | - | - | - | - | - | - | - | - | - | - |
| | REFERS | 84.7 | 83.0 | 92.3 | 82.1 | 90.2 | 88.7 | 91.4 | 83.9 | 93.3 | **74.1** | 85.5 | 76.7 | 78.5 | 77.0 | 89.1 |
| | Model Genesis | 81.0 | 78.8 | 84.5 | 79.2 | 87.8 | 86.6 | 89.7 | 81.0 | 85.2 | 71.1 | 81.9 | 73.2 | 75.8 | 73.0 | 85.6 |
| | C2L | 82.2 | 81.1 | 90.2 | 81.0 | 88.1 | 88.0 | 88.3 | 80.8 | 86.8 | 72.0 | 82.7 | 74.1 | 76.2 | 75.3 | 85.9 |
| | Context Restoration | 78.7 | 75.8 | 82.9 | 76.4 | 86.6 | 84.8 | 88.2 | 78.6 | 83.0 | 70.0 | 79.6 | 69.5 | 73.2 | 69.4 | 84.0 |
| | TransVW | 81.7 | 79.8 | 85.0 | 80.0 | 88.2 | 87.1 | 90.1 | 81.8 | 85.9 | 72.3 | 82.6 | 74.4 | 76.6 | 74.0 | 86.1 |
| | ImageNet Pre-training | 80.0 | 78.3 | 89.3 | 77.6 | 87.9 | 85.9 | 87.4 | 78.5 | 88.8 | 65.9 | 79.9 | 70.7 | 74.5 | 71.0 | 84.7 |

Table 2: COMPARISONS ON NIH CHESTX-RAY. Besides self-supervised and transfer learning baselines, we also present the performance of REFERS and MedKLIP (with competitive performance on CheXpert, RSNA Pneumonia, and SIIM) as references. AUC scores are displayed. The highest AUC scores in each labeling ratio are bolded.

**COVID-19 Image Data Collection** is a relatively small dataset, which involves 900 chest radiographs. We follow Zhou et al. (2022) to conduct fine-tuning on this small-scale dataset to investigate the effectiveness of various pre-training methodologies when the amount of annotations is limited. There are two tasks included. The first task requires the model to distinguish COVID-19 cases from non-COVID-19 pneumonia cases, where the training/validation/test set comprises 356/54/99 radiographs, respectively. The second task is to distinguish viral pneumonia cases from bacterial pneumonia ones, where the training/validation/test set contains 297/43/86 cases, respectively.

**SIIM-ACR Pneumothorax Segmentation (SIIM)** aims to facilitate the development of segmentation models to identify pneumothorax disease in chest radiographs. SIIM contains over 120,000 frontal-view chest X-rays with precise manual segmentation of pneumothorax. We follow Huang

et al. (2021) to construct the training/validation/test split, where each constitutes 70%/15%/15% of the whole dataset.

### 4.3 BASELINES

#### 4.3.1 REPORT-SUPERVISED METHODOLOGIES

We first compare MRM against a range of pre-training approaches, which use radiology reports as supervision to learn radiograph representations. There are 4 report-supervised approaches involved in the baseline comparisons, which are ConVIRT (Zhang et al., 2020), GLoRIA (Huang et al., 2021), BioViL (Boecking et al., 2022), and REFERS (Zhou et al., 2022). Specifically, *ConVIRT* (Zhang et al., 2020) proposed to learn medical visual representations by contrasting paired radiographs and sentences from radiology reports. *GLoRIA* (Huang et al., 2021) improved ConVIRT by contrasting radiograph sub-regions and words in the reports. *BioViL* (Boecking et al., 2022) and *REFERS* (Zhou et al., 2022) incorporated masked language modeling loss into contrastive learning. Moreover, REFERS introduced a multi-view fusion attention to better align the representations of each radiograph and its associated report. In addition, MGCA (Wang et al., 2023) and Med-KLIP (Wu et al., 2023) were included as two recent baselines[4]. Apart from above baselines, we also include M3AE (Geng et al., 2022), a recent masked multi-modal pre-training method aside from the application to medical data, for comparison.

In experiments, we fine-tune pre-trained models of MRM and other report-supervised methods on CheXpert (classification), RSNA Pneumonia (classification), and SIIM (segmentation).

#### 4.3.2 SELF-SUPERVISED AND TRANSFER LEARNING METHODS

Besides report-supervised approaches, we also include self-supervised and transfer learning approaches in our comparisons. Specifically, Context Restoration (Chen et al., 2019), Model Genesis (Zhou et al., 2021b), and TransVW (Haghighi et al., 2021) are based on predictive SSL, while C2L (Zhou et al., 2020) was developed on top of contrastive learning. In addition, MRM is also compared to ImageNet pre-training (Wang et al., 2017). In practice, we conduct the comparisons with self-supervised and transfer learning approaches on NIH ChestX-ray and COVID-19 Image Data Collection.

| Methods | COVID-19 vs. Others | Viral vs. Bacterial |
|---|---|---|
| Our MRM | **85.8**± 0.4 | **91.5**± 0.3 |
| REFERS | 82.1 | 80.4 |
| Model Genesis | 76.0 | 71.8 |
| C2L | 77.8 | 73.0 |
| Context Restoration | 74.6 | 69.8 |
| TransVW | 76.1 | 71.5 |
| ImageNet Pre-training | 74.1 | 70.3 |

| Methods | 1% | 10% | 100% |
|---|---|---|---|
| Our MRM | **79.4** | **84.0** | **85.9** |
| - Masked modeling & Super-resolution restoration | 69.9 | 75.2 | 80.3 |
| - Masked report modeling ($\mathcal{L}_R$) | 74.7 | 81.3 | 85.1 |
| - Masked radiograph modeling ($\mathcal{L}_I$) | 76.7 | 82.2 | 84.7 |
| - Super-resolution restoration | 78.8 | 83.7 | 85.7 |
| + Hybrid features for image restoration | 78.9 | 83.6 | 85.7 |

Table 3: COMPARISONS ON COVID-19 IMAGE DATA COLLECTION. The best results are bolded.

Table 4: ABLATIONS ON NIH CHESTX-RAY. AUC scores of three different labeling ratios are reported. The best results are bolded.

### 4.4 RESULTS

#### 4.4.1 COMPARISONS WITH REPORT-SUPERVISED BASELINES

In Table 1, we present the comparative results with report-supervised methodologies on CheXpert, RSNA Pneumonia, and SIIM (segmentation). Specifically, we investigate the performance when fine-tuning with limited and full supervision. We provide reconstruction examples and segmentation results in the appendix.

From Table 1, we observe no obvious performance gap between CNN- and Transformer-based report-supervised pre-training methods. For instance, after implementing GLoRIA on MIMIC-CXR with ViT-B/16, we observe performance drops and improvements on CheXpert and RSNA Pneumonia, respectively. This contrast demonstrates that replacing CNN with Transformer may not bring

---

[4]MGCA (Wang et al., 2023) and MedKLIP (Wu et al., 2023) were released after the submission deadline of ICLR 2023. We added results in the camera ready for better comparisons.

| Fusion | 1% | 10% | 100% |
|---|---|---|---|
| GAP | 79.4 | 84.0 | 85.9 |
| GMP | 77.2 | 83.8 | 86.0 |

| $\mathcal{P}_I$ | 1% | 10% | 100% |
|---|---|---|---|
| 75% | **79.4** | 84.0 | 85.9 |
| 50% | 78.8 | **84.4** | **86.1** |
| 0% | 75.4 | 82.0 | 84.4 |

| $\lambda$ | 1% | 10% | 100% |
|---|---|---|---|
| 1 | **79.4** | **84.0** | **85.9** |
| 2 | 78.6 | 83.5 | 85.4 |
| 0.5 | 79.0 | 83.7 | **85.9** |

(a) Multi-modal fusion strategies  (b) Masking ratios for radiographs  (c) Loss controlling factor $\lambda$

Table 5: Ablation studies on choices of multi-modal fusion and hyper-parameters. **GAP** and **GMP** stand for global average and maximum pooling, respectively. Experiments are performed on NIH ChestX-ray under various labeling ratios. The best results are bolded.

performance gains to report-supervised pre-training. Among all baselines, REFERS is the best performing approach.

Nonetheless, MRM consistently outperforms various baselines on all three datasets under different labeling ratios. Specifically, MRM maintains more advantages over previous pre-training methodologies when fine-tuning with limited annotations, which is quite meaningful for medical image analysis as large amounts of specialist annotations (from radiologists or clinicians) are usually hard to access. *It is worth noting that MRM achieves 88.5% when using only 1% labeled data on CheXpert, better than previous counterparts with 100% annotations.*

We see that M3AE generally performs worse than our MRM in all labeling ratios, especially under extremely limited data. The underperformance of M3AE may be attributed to the fact that it requires a large amount of multi-modal data to learn transferable joint representations for images and texts.

### 4.4.2 COMPARISONS WITH SELF-SUPERVISED AND TRANSFER LEARNING BASELINES

Table 2 presents the average and per-class classification results on NIH ChestX-ray. Compared to self-supervised learning baselines, MRM achieves large improvements in almost every chest pathology. On average, MRM outperforms C2L (Zhou et al., 2020) and TransVW (Haghighi et al., 2021), the best two self-supervised pre-training methodologies, by about 8% when the amount of available labeled data is extremely limited (i.e., the labeling ratio is 1%). Similarly, we also observe remarkable improvements when comparing MRM to ImageNet Pre-training (Wang et al., 2017). These phenomena demonstrate that the radiograph representations learned by MRM are more transferable than those from previous self-supervised and transfer learning methods.

Compared to REFERS (i.e., the best performing report-supervised baseline in Table 1), MRM still provides notable improvements in most chest pathologies. Specifically, MRM is more advantageous when the amount of labeled data is limited. For instance, MRM surpasses REFERS by 2.7% and 3.1% on average when the labeling ratios are 1% and 10%, respectively. These comparisons again verify the effectiveness of MRM over previous report-supervised pre-training counterparts.

We also investigate the impacts of pre-training on the real-world scenario with extremely limited specialist supervision. In Table 3, we compare the performance of a range of pre-training methodologies on two binary classification tasks, which are distinguishing COVID-19 from non-COVID-19 pneumonia, and differentiating between viral pneumonia and bacterial pneumonia. Again, MRM outperforms self-supervised, transfer learning, and report-supervised pre-training methodologies by substantial margins. Compared to REFERS, MRM brings nearly 4% and 11% improvements to two tasks, respectively, further enhancing the practicability of the diagnosis system trained with extremely limited supervision.

### 4.5 ABLATION ANALYSIS

**Advantages over single-task pre-training paradigm.** First of all, we remove the two masked modeling objectives and super-resolution restoration task, resulting in substantial performance drops. These results verify the necessity of using masked modeling and super-resolution restoration in MRM. After removing the masked report modeling objective, MRM only acquires supervision signals from self-supervision. Thus, the whole framework degenerates into a self-supervised pre-training methodology. From Table 4, we observe that removing $\mathcal{L}_R$ leads to dramatic performance degradation in different labeling ratios. Moreover, we find that introducing the masked report mod-

eling is greatly helpful to the fine-tuning performance with limited labeling resources (1% and 10%). For instance, adding $\mathcal{L}_\mathrm{R}$ brings about 5-percent improvements to MRM. Removing the masked radiograph modeling also leads to notable performance drops in all labeling ratios.

**Is super-resolution restoration helpful?** As Table 4 shows, the proposed super-resolution restoration provides consistent performance gains in different labeling ratios. The underlying reason may be that the low to high resolution restoration process helps preserve more local information into latent representations, which enhances the transferable ability to downstream tasks.

**Would it be beneficial to introduce multi-modal information to image restoration?** We investigate this question by adding non-masked report token embeddings to image patch embeddings and passing the resulting hybrid features to the image decoder. As Table 4 shows, introducing multi-modal information to masked image restoration does not improve the fine-tuning performance. We leave the exploration of the reason behind to future work.

**Ablations on multi-modal fusion and hyper-parameters.** In Table 5a, we find that global average pooling (GAP) outperforms global maximum pooling (GMP) by 2.2% when access to labeled data is quite limited while achieving comparable results as the labeling ratio increases. We also investigate the impact of applying different masking ratios to input radiographs (cf. Table 5b). Specifically, we find that a ratio of 75% performs the best on the extremely small labeling ratio (i.e., 1%), while a ratio of 50% achieves slightly better results when the labeling ratio becomes larger. In MRM, we set the default masking ratio for radiographs to 75% because this operation leads to fewer input patches, accelerating the pre-training process and reducing the memory cost. The insight behind applying a high masking ratio (i.e., 75%) is that it addresses the heavy spatial redundancy of radiographs. We also perform ablative experiments to investigate the influence of $\lambda$ in Eq. 3, whose results are displayed in Table 5c. We see that $\lambda = 1$ is an optimal choice, while a smaller $\lambda$ value (i.e, 0.5) performs better than a larger value (i.e., 2.0), indicating that the MLM objective may play a more important role than the MIM objective during the pre-training stage.

## 4.6 IMPLEMENTATION DETAILS.

Our code is implemented using PyTorch 1.8.2 (Paszke et al., 2019). The pre-training experiments were conducted on 4 GeForce RTX 3080Ti GPUs, and the training time is about 2 days for 200 epochs, requiring 12GB memory from each GPU. The training batch size is 256. We use AdamW (Loshchilov & Hutter, 2017) as the default optimizer, where the initial learning rate is $1.5\mathrm{e}^{-4}$, weight decay is 0.05, $\beta_1$ is 0.9, and $\beta_2$ is 0.95. The MSE and cross-entropy losses are used for masked image and language modeling, respectively. In practice, we set $\lambda$ in Eq. 3 to 1.

For fine-tuning on SIIM, we train the segmentation network on 4 GeForce RTX 3080Ti GPUs. AdamW is the default optimizer, where the initial learning rate is $2\mathrm{e}^{-5}$, weight decay is 0.05, $\beta_1$ is 0.9, and $\beta_2$ is 0.999. For fine-tuning on other datasets, we train the classification network on a single GeForce RTX 3080Ti GPU, where the default optimizer is SGD with momentum 0.9. The training cost is a mix of focal and dice losses.

For fine-tuning on CheXpert, RSNA Pneumonia, NIH ChestX-ray, and COVID-19 Image Data Collection, we adopt the cross-entropy loss. We search the best initial learning rate from 3e-2, 3e-3, and 5e-4 to get the best performance on validation set.

For both the pre-training and the fine-tuning of image classification task, the network is "warmed up" by increasing the learning rate linearly to the set value, and then learning rate is decreased using the cosine decay schedule.

## 5 CONCLUSION

We present a unified yet simple framework based on masked record modeling (MRM) for radiograph representation learning. MRM formalizes the radiograph understanding and radiology report comprehension as two complementary masked modeling objectives. In practice, MRM learns pre-trained models that generalize well. With MRM pre-training, we achieve better results on well-established datasets. Specifically, MRM outperforms previous self- and report-supervised counterparts by large margins when the labeled data is extremely limited.

ACKNOWLEDGEMENTS

This work was supported by Hong Kong Research Grants Council through General Research Fund (Grant 17207722) and the National Key Research and Development Program of China (Grant STI2030-Major Projects2021ZD0201900).

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

# A   APPENDIX

## A.1   DISCUSSION: DIFFERENCES FROM MASKED VISION-AND-LANGUAGE PRE-TRAINING BESIDES MEDICAL DATA

To our knowledge, M3AE (Geng et al., 2022), MLIM (Arici et al., 2021), and MaskVLM (Kwon et al., 2022) are three recent efforts that are most closely related to ours, all of which use masked modeling for vision and language pre-training. In the following, we clarify the differences between our MRM and these approaches from two perspectives: motivation and implementation.

**Motivation**. M3AE (Geng et al., 2022) and MLIM (Arici et al., 2021) aim to learn joint image-text representations. MaskVLM (Kwon et al., 2022) is developed for improving vision+language tasks, such as image-text retrieval, visual question answering, and natural language for visual reasoning. These characteristics differ from our methodology, which aims to learn a representation for radiographs only for disease diagnosis even though both radiographs and reports are used during training.

**Implementation**. As aforementioned, M3AE (Geng et al., 2022) and MLIM (Arici et al., 2021) implement unified multi-modal transformers that take imaging and textual as inputs. As a result, they require much more pre-training data, which is often an order of magnitude larger than ours. This is not practical and applicable in the medical field, where access to the data is quite limited. MaskVLM (Kwon et al., 2022) applies masked modeling and contrastive learning to image-text pairs, where a binary matching task is employed to tell whether an image and a text are aligned or not. Besides, MaskVLM builds cross-modality encoders to incorporate multi-modal information. In contrast, our MRM is much simpler. MRM uses masking modeling as the only training objective, and a simple GAP-addition workflow is proposed for fusing multi-modal information.

## A.2   CONFIGURATIONS OF NETWORKS

The image encoder a ViT-like encoder (Dosovitskiy et al., 2020) that includes a patch embedding layer followed by twelve transformer blocks. The architecture of the image decoder is very similar to that of the encoder. The decoder architecture consists of a decoder embedding layer, four transformer blocks, and one fully-connected layer to predict masked patches. The numbers of attention heads in the image encoder and decoder are twelve and six, respectively. We add a 2D sin-cos positional embedding to input patches. The report decoder is a light-weight transformer that includes an embedding layer and six transformer blocks, followed by a one-layer fully-connected predictor. The number of attention heads in the report decoder is six. We adopt a learnable positional embedding in the report decoder.

## A.3   RECONSTRUCTION ANALYSIS

Fig. 3 presents example results on MIMIC-CXR. We find that even with high masking ratios, MRM can still produce satisfactory reconstructions, though some details are missing. Specifically, the reconstructed reports are surprisingly close to the ground truths, which we attribute to the introduction of hybrid multi-modal representations. For instance, MRM can tell the position of rib fractures based on the radiograph input. Besides, we obtain some interesting mistakes. In the first example, MRM recognizes the clips over the left lung as calcifications (potentially in nipple shadow). This observation again shows that the report reconstructions rely on the image input as the clips are masked in the input radiograph.

## A.4   SEGMENTATION ANALYSIS

In this section, we visualize the segmentation results of GLoRIA (Huang et al., 2021), REFERS (Zhou et al., 2022), and our MRM.

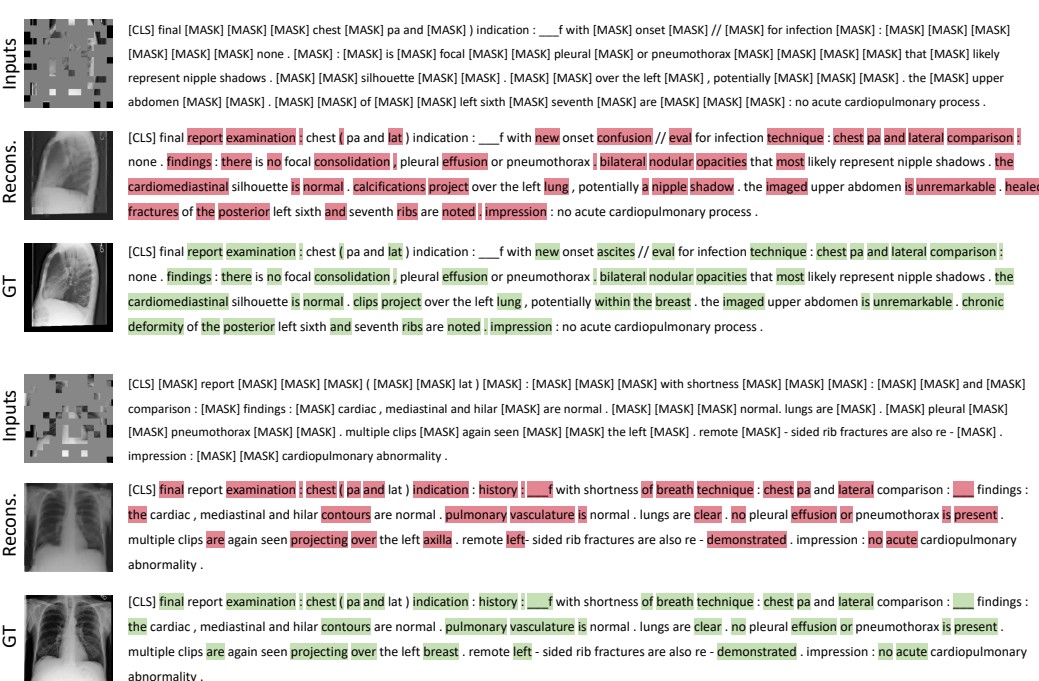

Figure 3: Example results on MIMIC-CXR. For each triplet, we show the masked radiograph and report (Inputs), our MRM reconstruction (Recons.), and the ground truth (GT). The masking ratios are 75% (radiograph) and 50% (report). Predicted and corresponding ground truth words are highlighted in pink and green, respectively.

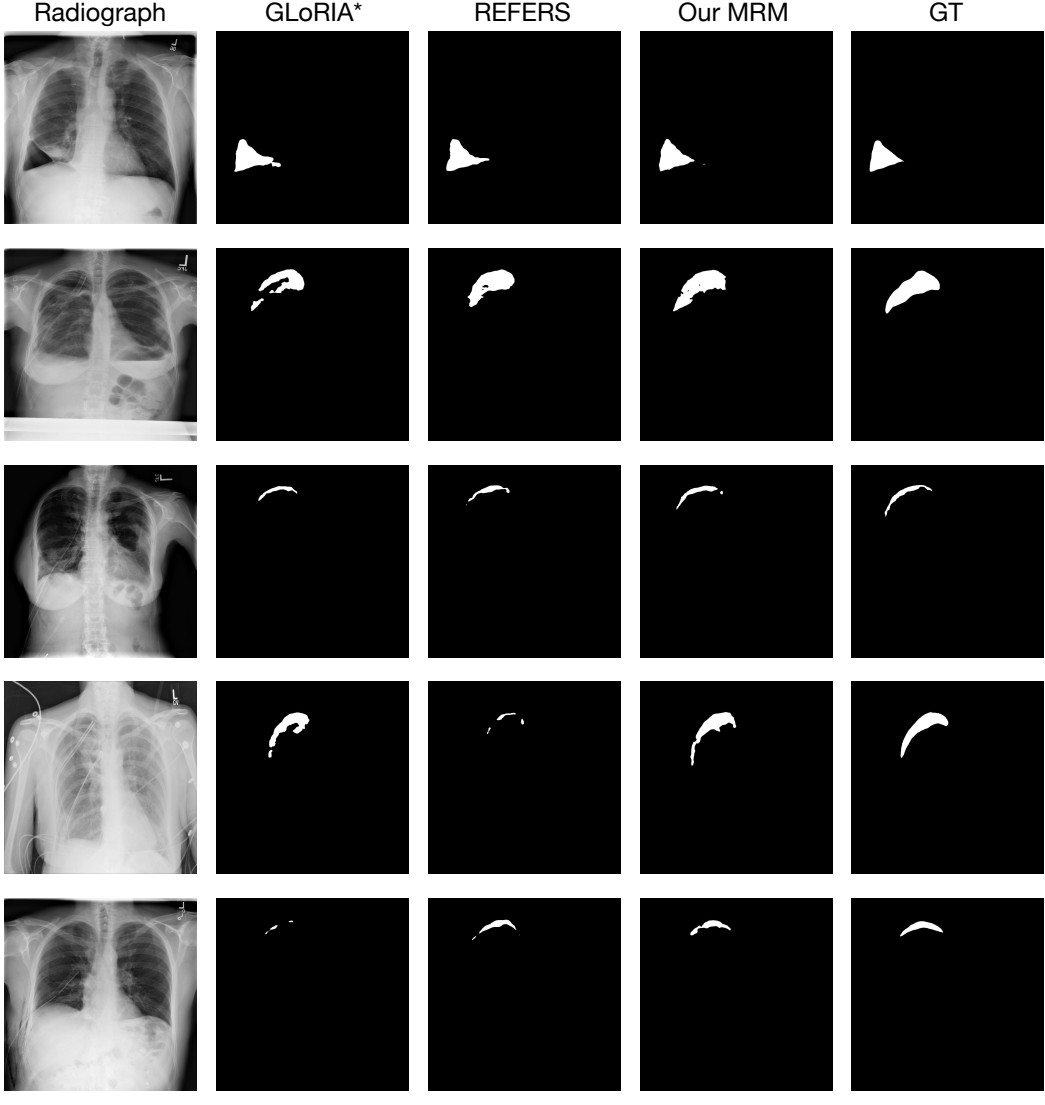

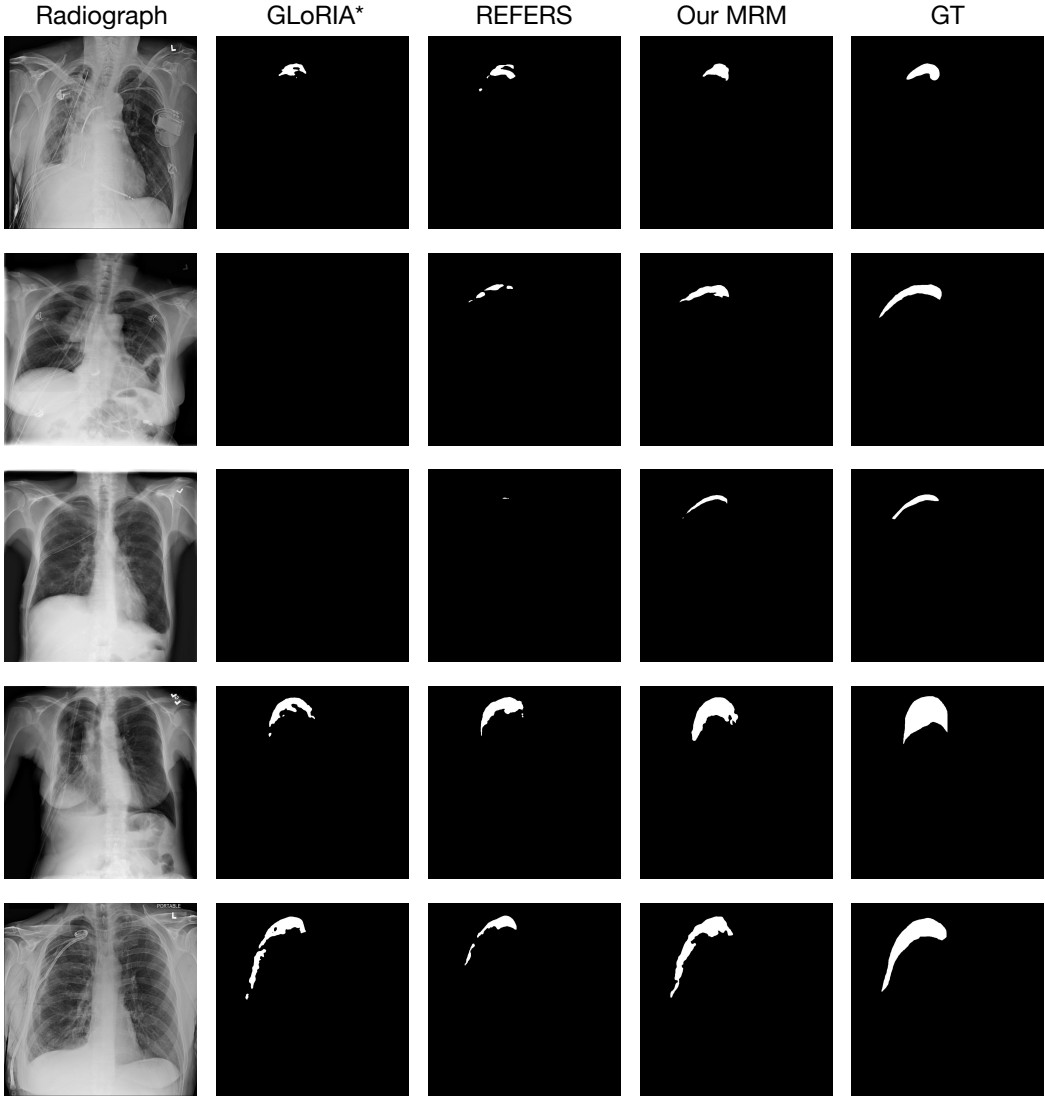

Figure 4: Segmentation results of MRM, GLoRIA, and REFERS. **GT** stands for the ground truth masks. * means the GLoRIA implementation is based on ViT-B/16, the same backbone as used in REFERS and MRM.

