# OpenReview forum: "Advancing Radiograph Representation Learning with Masked Record Modeling"
_ICLR.cc/2023/Conference — ICLR 2023 poster_

### Official Review · Reviewer_35Lo · 2022-10-24

**Confidence:** 4
**Correctness:** 3
**Technical Novelty And Significance:** 3
**Empirical Novelty And Significance:** 3
**Recommendation:** 8

**Clarity, Quality, Novelty And Reproducibility:**

This paper lacks certain clarity on its own, as mentioned above. Some technical details are missing. However, the authors provided their source code. If the missing details are completed, it should be possible to reproduce the results.

The idea of combining masked image and language modeling is not entirely novel but may be so for radiograph representation learning. There are previous efforts on joint masked image and language modeling [1,2]. However, they are not published yet.
[1] MLIM: Vision-and-Language Model Pre-training with Masked Language and Image Modeling, https://arxiv.org/abs/2109.12178
[2] Masked Vision and Language Modeling for Multi-modal Representation Learning, https://arxiv.org/abs/2208.02131

The quality of this work is high as it made a large improvement to radiograph representation learning compared to previous methods.

**Strength And Weaknesses:**

Strength

- The paper is well-written and easy to follow. The proposed method is explained well with text descriptions and figure illustrations.
- The proposed method, Masked Record Modeling (MRM), combining masked image and language modeling, is effective for learning radiograph representations. MRM outperforms state-of-the-art both report-supervised and self-supervised / transfer learning methods by large margins in some cases.
- The ablation study shows the contributions of individual components in the proposed method. Specifically, it shows masked language modeling has a significant impact on improving radiograph representation learning.

Weakness

- Some technical details need more clarification. What is the lookup table for report token embedding? What are the specific configurations of the image encoder, image decoder, and report decoder?
- In Table 2, MRM outperforms REFERS in most categories, except for “Infiltration”. Why? It is worth conducting a more in-depth analysis.
- In Section 4.4, “The cross-entropy and MSE losses are used for masked image and language modeling, respectively.” The order of two losses should be swapped.

**Summary Of The Paper:**

This paper presents a new approach to learning radiograph representation through a joint masked image and language modeling. Specifically, for each pair of radiograph and radiology report pair, both the image and the report are masked for restoration tasks. The masked image patches are encoded and used for both restorations, while the masked text embedding is only used for report restoration. Besides, the input image is downsampled which combines a super-resolution task. The model is pre-trained on the large MIMIC-CXR dataset and then fine-tuned for downstream tasks: classification and segmentation on CheXpert, NIH ChestX-ray, RSNA Pneumonia, COVID-19 Image Data Collection, and SIIM-ACR Pneumonia Segmentation datasets. Experiment results show the proposed method outperforms all comparison methods, including report-supervised, self-supervised, and transfer learning methods. Ablation study experiments also provide an analysis of contributions from individual components in the proposed method.

**Summary Of The Review:**

This paper combined masked image and language modeling for radiograph representation learning and showed significant improvement over existing methods for downstream tasks. The weaknesses of this paper could be addressed with some more effort.

---

> ### Author Response · Authors · 2022-11-16
> **Response to Reviewer 35Lo (PART1)**
>
> Thanks for your positive comments about the novelty, experiments, and paper writing. In the following, we address your concerns by providing point-by-point responses. Meanwhile, if you have further questions or problems, please let us know.
>
> ------------------------------------
> **Comment-1**:
> > Some technical details need more clarification. What is the lookup table for report token embedding? What are the specific configurations of the image encoder, image decoder, and report decoder?
>
> **Response-1**:
>
> Thanks for your kind comments. We would like to clarify these details as follows (and codes and models will be released). These clarifications have been added to the updated manuscript.
> - Lookup table: This is an existing function in PyTorch (cf. [torch.nn.Embedding](https://pytorch.org/docs/stable/generated/torch.nn.Embedding.html)). We use this function to create a collection of randomly initialized embeddings. The number of these embeddings is equal to the number of tokens so that each token corresponds to a specific randomly initialized embedding, and we can retrieve the embedding given the index of a token.
>
> - Image encoder: The image encoder is a ViT-like encoder (Dosovitskiy et al., 2020) that includes a patch embedding layer followed by twelve transformer blocks.
>
> - Image decoder: The architecture of the image decoder is very similar to that of the encoder. The decoder includes a decoder embedding layer, four transformer blocks, and one fully-connected layer to predict masked patches.
>
> Specific configurations of the image encoder and decoder are given as follows:
> ||Image encoder|Image decoder|
> |----|----|----|
> Input size|224x224|N/A|
> Patch size|16x16|N/A|
> Embedding dimension|768|768|
> Num. of transformer blocks|12|4|
> Num. of attention heads|12|6|
> Output size|N/A|448x448|
> Positional Embedding|2D sin-cos|2D sin-cos|
>
> - Report decoder: The report decoder is a light-weight transformer that includes an embedding layer and six transformer blocks, followed by a one-layer fully-connected predictor.
>
> Specific configurations are as follows:
> ||Report decoder|
> |----|----|
> Max input length|100|
> Vocabulary size|30,000|
> Embedding dimension|384|
> Num. of transformer blocks|6|
> Num. of attention heads|6|
> Positional Embedding|Learnable|
>
> ------------------------------------
> **Comment-2**:
> > In Table 2, MRM outperforms REFERS in most categories, except for "Infiltration". Why? It is worth conducting a more in-depth analysis.
>
> **Response-2**:
>
> We appreciate your carefulness in reviewing our paper; this is an insightful comment. Infiltration is the worst-performing pathology of all approaches, as shown in Table 2 in the manuscript. One methodological difference between REFERS and our MRM is that REFERS adopts an auto-regressive way to learn from radiology reports (i.e., learning to generate reports). The success of REFERS on infiltration can be partly attributed to the auto-regressive pre-training manner, which may help encode more relevant information into visual representations. We will continue to explore this issue in our future work.
>
> ------------------------------------
> **Comment-3**:
>
> > In Section 4.4, “The cross-entropy and MSE losses are used for masked image and language modeling, respectively.” The order of two losses should be swapped.
>
> **Response-3**:
>
> Thanks for pointing out this typo. We have fixed it in our revised manuscript.

---

> > ### Author Response · Authors · 2022-11-16
> > **Response to Reviewer 35Lo (PART2)**
> >
> > **Comment-4**:
> >
> > > There are previous efforts on joint masked image and language modeling [1,2]. However, they are not published yet. [1] MLIM: Vision-and-Language Model Pre-training with Masked Language and Image Modeling, https://arxiv.org/abs/2109.12178 [2] Masked Vision and Language Modeling for Multi-modal Representation Learning, https://arxiv.org/abs/2208.02131.
> >
> > **Response-4**:
> >
> > Thank you for listing these two relevant works. We have added them to the related work. Besides, we have added a discussion section in the appendix, where we clarify the differences between our methodology and recent efforts (including [1,2]) on masked vision-and-language pre-training (aside from medical data) from two perspectives: motivation and implementation. We provide an excerpt here for your consideration:
> >
> > - **Motivation**. MLIM [1] aims to learn joint image-text representations. MaskVLM [2] is developed for improving vision+language tasks, such as image-text retrieval, visual question answering, and natural language for visual reasoning. These characteristics differ from our methodology, which aims to learn a representation for radiographs only for disease diagnosis even though both radiographs and reports are used during training.
> >
> > - **Implementation**. MLIM [1] implements a unified multi-modal transformer that takes imaging and textual as inputs. As a result, MLIM requires much more pre-training data, an order of magnitude larger than ours. This is not practical and applicable in the medical field, where access to the data is quite limited. MaskVLM [2] applies masked modeling and contrastive learning to image-text pairs, where a binary matching task is employed to determine whether an image and a text are aligned. Besides, MaskVLM builds cross-modality encoders to incorporate multi-modal information. In contrast, our MRM is much simpler. MRM uses masking modeling as the only training objective, and a simple GAP-addition workflow is proposed for fusing multi-modal information.
> >
> > ------------------------------------
> >
> > Again, we sincerely appreciate your time in reviewing our paper. According to your comments and advice, we have carefully revised our manuscript by adding more implementation details and discussion. If you have further questions or suggestions, please do let us know.

---

> > > ### Comment · Reviewer_35Lo · 2022-11-30
> > > **Thank you for the responses**
> > >
> > > Thank you for addressing my concerns! I have no further questions and maintain my original rating.

---

> ### Author Response · Authors · 2022-11-18
> **Follow up**
>
> Dear Reviewer 35Lo,
>
> Again, thank you for your time and efforts in reviewing our paper and providing constructive feedback. We hope our response has appropriately addressed your concerns.
>
> Thanks for your positive feedback. They do encourage us a lot! We will continue to explore this topic in future work.
>
> Paper1629 Authors

---

### Official Review · Reviewer_JZho · 2022-10-24

**Confidence:** 4
**Correctness:** 4
**Technical Novelty And Significance:** 3
**Empirical Novelty And Significance:** 3
**Recommendation:** 6

**Clarity, Quality, Novelty And Reproducibility:**

The method is clearly explained and novel. They have shared the code to reproduce the results.

**Strength And Weaknesses:**

Strength(s):
1. The method is novel and explained well.
2. The method outperforms other methods by a considerable margin on multiple datasets thereby signifying the method efficiently utilizes the signals from reports to learn better image embeddings.
3. The ablation experiments of different components nicely demonstrate how each component contributes to the performance.

Weakness(es)/Suggestion(s):
1. It would be nice to have confidence intervals/ SD around the metrics used to compare different methods.
2. Why GAP was used to share image information with the masked record decoder? What happens if we use other alternatives like Max pooling, or have a transfer module that takes in embedded patches and returns a global embedding for the image?
3. How is the lambda parameter in equation 3 decided?
4. Some ablation studies to investigate how well the model learns for different masking ratios and masking patterns would be interesting for the readers.

**Summary Of The Paper:**

The paper discusses a novel method to enhance the pre-train chest x-ray image embeddings using the associated report. They train two masked auto-encoders (one for reports and the other for images). They then combine the image patch embeddings with report embeddings to decode the masked report tokens. At the same, they also decode a high-resolution image using the same image patch embeddings. They compare their method with existing methods and show that they outperform other methods by a considerable margin on multiple datasets.

**Summary Of The Review:**

In summary, I found the method introduced in the paper to be interesting, novel, and elegant. The results across multiple datasets and downstream tasks demonstrate that the method is better than other methods at learning insights from reports to improve image embeddings. There are some important ablation experiments missing from the paper which would only strengthen the paper. I, therefore, give a recommendation of marginally above the acceptance threshold.

---

> ### Author Response · Authors · 2022-11-16
> **Response to Reviewer JZho**
>
> We appreciate your kind comments and insightful suggestions, which are greatly helpful to the revision of the manuscript. Specifically, we would like to thank you for pointing out promising directions, e.g., improving the multi-modal fusion module and investigating more masking patterns. In the following, we give point-by-point responses to address your concerns. We have carefully revised the manuscript according to your advice. Please do let us know if you have any other questions.
>
> ---------------------------------------
> **Comment-1**:
>
> > It would be nice to have confidence intervals/SD around the metrics used to compare different methods.
>
> **Response-1**:
>
> Thanks for your insightful advice. We have added standard deviations (SD) to the updated manuscript. Since some experimental results are directly quoted from their associated papers that did not report SDs, we provide the SDs from our implementations.
>
> ---------------------------------------
> **Comment-2**:
> > Why GAP was used to share image information with the masked record decoder? What happens if we use other alternatives like Max pooling, or have a transfer module that takes in embedded patches and returns a global embedding for the image?
>
> **Response-2**:
>
> This is an insightful comment. We have conducted an ablation study that compares GAP (global average pooling) with GMP (Global Maximum Pooling), whose results are shown in Table 5(a) in the updated manuscript.
> For your convenience, we provide an excerpt here:
>
> |Fusion Method|1%|10%|100%|
> |----|----|----|----|
> |GAP|79.4|84.0|85.9|
> |GMP|77.2|83.8|86.0|
>
> The above table shows that global average pooling (GAP) outperforms global maximum pooling (GMP) by 2.2% when access to labeled data is quite limited while achieving comparable results as the labeling ratio increases. The insight behind this may be that GAP encodes more contextual information into representations, preventing overfitting when a pretrained model is fine-tuned with limited data.
>
> We have yet to perform a thorough investigation on how to incorporate image information into the report side in a better way (e.g., using attention modules). We would like to thank you for pointing out a promising direction for improving our approach. We will continue to investigate this topic in our future work.
>
> ---------------------------------------
> **Comment-3**:
> > How is the lambda parameter in equation 3 decided?
>
> **Response-3**:
>
> Thanks for your comment. We have added an ablation study of the lambda parameter in Eq. 3 to the updated manuscript, and the experimental results are presented in Table 5(c). We provide an excerpt here for your consideration:
>
> |$\lambda$|1%|10%|100%|
> |----|----|----|----|
> |1|79.4|84.0|85.9|
> |2|78.6|83.5|85.4|
> |0.5|79.0|83.7|85.9|
>
> The above results show that $\lambda=1$ is an optimal choice. In contrast, a smaller $\lambda$ value (i.e., 0.5) performs better than a larger value (i.e., 2.0), indicating that the MLM objective may play a more critical role than the MIM objective during the pre-training stage.
>
> ---------------------------------------
> **Comment-4**:
> > Some ablation studies to investigate how well the model learns for different masking ratios and masking patterns would be interesting for the readers.
>
> **Response-4**:
>
> We appreciate your advice. The revised draft has included an ablation study of different masking ratios (please refer to Table 5(b)). We provide an excerpt here for your consideration：
>
> |$\mathcal{P}_I$|1%|10%|100%|
> |----|----|----|----|
> |75%|79.4|84.0|85.9|
> |50%|78.8|84.4|86.1|
> |0%|75.4|82.0|84.4|
>
> From the above table (all experiments are performed on NIH ChestX-ray), we find that a ratio of 75\% performs the best on the minimal labeling ratio (i.e., 1\%). In contrast, a ratio of 50\% achieves slightly better results as the labeling ratio increases. In MRM, we set the default masking ratio for radiographs to 75\% because this operation leads to fewer input patches, accelerating the pre-training process and reducing the memory cost.
>
> We leave the study of different masking patterns, such as applying dropout to high-confidence patches on the basis of the feedback from heatmaps, to future work.
>
> ---------------------------------------
>
> Again, we are grateful for your positive comments regarding the technical novelty, performance improvements, and paper writing. We hope our responses address your concerns. If you have further questions or suggestions, please do not hesitate to post them.

---

> > ### Comment · Reviewer_JZho · 2022-11-18
> > **Thank you for the responses.**
> >
> > Thank you authors for the responses. The authors have addressed my concerns. I will keep my score of 6 (marginally above the acceptance threshold) unchanged.

---

> > > ### Author Response · Authors · 2022-11-18
> > > **Reply to Reviewer JZho**
> > >
> > > Dear Reviewer JZho,
> > >
> > > Thank you very much for your helpful reviews and your supportive comments! We are happy that our revision has addressed your concerns. We sincerely appreciate your constructive suggestions.
> > >
> > > Thank you!
> > >
> > > Paper1629 Authors

---

> ### Author Response · Authors · 2022-11-18
> **Follow up**
>
> Dear Reviewer JZho,
>
> We sincerely appreciate your efforts in provding insightful comments and constructive advice. We hope our response has appropriately addressed your concerns. If you still feel unclear or concerned, please let us know, and we will be more than glad to clarify and discuss any further concerns. If you feel your concerns have been addressed, please kindly consider if it is possible to update your score.
>
> Thank you!
>
> Paper1629 Authors

---

### Official Review · Reviewer_Xf1P · 2022-10-25

**Confidence:** 3
**Correctness:** 3
**Technical Novelty And Significance:** 2
**Empirical Novelty And Significance:** 2
**Recommendation:** 6

**Clarity, Quality, Novelty And Reproducibility:**

Overall, the paper is very well written and easy to understand. One part that is missing is a discussion of whether multi-modal self-supervised learning has been tackled in other methods, potentially outside of medical imaging.
Also, what is the labelling ratio defined as?

The approach seems like a novel combination of existing image-based and text-based methods. Code is provided and most of the implementation is well described in the manuscript. It is not clear why the weight (lambda) of the report and the image loss is equal, have the authors experimented with this weight?


**Strength And Weaknesses:**

Strengths
- Paper is well written and easy to understand
- A thorough review of literature is provided
- Extensive experiments validate the proposed approach on the MIMIC-CXR (pre-training) and CheXPert, RSNA Pneumonia, NIH-Chest X-ray and COVID-19 Image Data Collection datasets (fine-tuning).
- Compelling results are presented showing the computational improvements in AUC with the proposed method

Weaknesses
- Not clear why masking is necessary for images, or whether the task could be super-resolution only
- Ablation studies with non-masking, non-super resolution pre-text tasks on records are missing
- Experimental comparisons to other multimodal (language+image) methods, such as M3AE (Geng et al. 2022) are missing


**Summary Of The Paper:**

The paper proposes a novel approach for self-supervised learning from radiographs and associated medical records. The proposed multimodal method learns to predict masked inputs from both record (test) and radiograph (image) data.


**Summary Of The Review:**

The paper seems like a straightforward combination of existing techniques, but generates compelling results on a variety of benchmarks. Novelty of combining MAE with BERT is already proposed by some papers, e.g. M3AE, and there is no clear and demonstrated improvement over these works, aside from the novel application to medical data. However, ablation experiments, discussion, and writing is clear and complete.

---

> ### Author Response · Authors · 2022-11-16
> **Response to Reviewer Xf1P (PART 1)**
>
> We sincerely appreciate your reception of our experimental results (compelling) and paper writing (very well written). We notice that your main concerns include a comparison to multi-modal (image+language) pre-training methods (e.g., M3AE) developed for natural images & language and the need for additional ablation studies. In the following, we address your concerns by including point-by-point responses. If you have further questions, please do let us know.
>
> ---------------------------------------------
> **Comment-1**:
>
> > Not clear why masking is necessary for images, or whether the task could be super-resolution only.
>
> **Response-1**:
>
> Thanks for your comments. An intuitive explanation for the necessity of masking is that masked image modeling incorporates more contextual information into learned image representations because reconstructing masked image patches requires non-masked patch embeddings to refer to the global context. Similar explanations have been adopted in BERT (Devlin et al., 2018) and MAE (He et al., 2022).
>
> We have conducted an ablation study to investigate the impact of applying different masking ratios to radiographs (the super-resolution task keeps unchanged). The experimental results are reported in Table 5(b) in the revised manuscript, and the associated analyses lie in the last paragraph of Sec. 4.5. We provide an excerpt here for your consideration:
>
> |$\mathcal{P}_I$|1%|10%|100%|
> |----|----|----|----|
> |75%|79.4|84.0|85.9|
> |50%|78.8|84.4|86.1|
> |0%|75.4|82.0|84.4|
>
> From the above table (all experiments are performed on NIH ChestX-ray), we find that a ratio of 75\% performs the best on the minimal labeling ratio (i.e., 1\%). In contrast, a ratio of 50\% achieves slightly better results as the labeling ratio increases. In MRM, we set the default masking ratio for radiographs to 75\% because this operation leads to fewer input patches, accelerating the pre-training process and reducing the memory cost.
>
> The insight behind applying a high masking ratio (i.e., 75%) is that it exploits the heavy spatial redundancy of radiographs. By applying a high masking ratio, we reduce the redundancy and create a surrogate task, requiring the model to understand the high-level semantics holistically.
>
> ---------------------------------------------
> **Comment-2**:
>
> > Ablation studies with non-masking, non-super resolution pre-text tasks on records are missing.
>
> **Response-2**:
>
> Thanks for your insightful and kind comment. In the updated manuscript, we include two additional ablation studies (in the 3rd and 6th rows) in Table 4 to address your concerns. We provide an excerpt here for your consideration (all experiments are performed on NIH ChestX-ray):
> |Methods|1%|10%|100%|
> |----|----|----|----|
> |MRM|79.4|84.0|85.9|
> |- Masked modeling & Super-resolution restoration|69.9|75.2|80.3|
> |- Super-resolution restoration|78.8|83.7|85.7|
>
> The above table shows that removing both masked modeling and super-resolution restoration tasks leads to substantial performance drops. These results verify the necessity of using masked modeling and super-resolution restoration in our approach. Besides, the proposed super-resolution restoration task gives rise to consistent performance gains under different labeling ratios. The underlying reason may be that the low to high-resolution restoration process helps preserve more local information in latent representations, which become more transferable to downstream tasks.
>
> ---------------------------------------------
> **Comment-3**:
>
> > One part that is missing is a discussion of whether multi-modal self-supervised learning has been tackled in other methods, potentially outside of medical imaging.
>
> **Response-3**:
>
> We appreciate your comments. In Sec. 2.2 (related work), we have discussed the relation to visual representation learning via image-language (multi-modal) pre-training. In the revised manuscript, we clarify the differences between our method and M3AE (Geng et al., 2022).
>
> Due to limited page space, we are afraid that adding more discussion about multi-modal self-supervised learning to the main text may not be possible. To address this concern, we add a part in the appendix, discussing potential connections to existing multi-modal self-supervised learning methodologies in detail.

---

> > ### Author Response · Authors · 2022-11-16
> > **Response to Reviewer Xf1P (PART 2)**
> >
> > **Comment-4**:
> > > Experimental comparisons to other multimodal (language+image) methods, such as M3AE (Geng et al. 2022), are missing.
> >
> > **Response-4**:
> >
> > Thanks for your comment. We would like to clarify that our work was completed independently and concurrently with M3AE (Geng et al., 2022). M3AE was first time uploaded to arXiv on 27 May 2022. However, we had already completed the algorithm part of our MRM on 3 May 2022 (cf. the [github commit history](https://github.com/DopamineLcy/MAE/commit/339374aceeedc19260a6e07ad6b2cd785ca289fd)).
> >
> > M3AE and our MRM are developed out of different motivations. According to the authors of M3AE:
> > > M3AE is a simple and scalable network architecture that learns a **unified encoder for both vision and language data** via masked token prediction.
> >
> > The motivation of M3AE is to learn a unified, general encoder for both vision and language. Thus, M3AE passes both images and texts to a shared encoder to *learn joint embeddings*. However, our goal is to *learn a specialized representation for radiographs only*. Based on this consideration, we disentangle multi-modal inputs. Our encoder only takes image patches as inputs, and we build an independent decoder for masked language modeling. Such a design has one potential merit, i.e., we can pre-train a performant transformer with fewer data. This is crucial for the medical field, where access to data and associated expertise is often limited. In contrast, training a multi-modal transformer often requires much more pre-training data. For instance, M3AE (Geng et al., 2022) uses 12M image-text items for pre-training, and MLIM (Arici et al., 2021) uses 6M items from Amazon catalog data for pre-training.
> >
> > Nonetheless, we agree that adding a multi-modal pre-training method aside from methods for medical data would be helpful for comparison. Based on this consideration, we choose M3AE as a representative baseline and add its experimental results to the revised Table 1.
> >
> > In the following table, we compare the results of our MRM and M3AE:
> > |Methods|C (1%)|C (10%)|C (100%)|P (1%)| P (10%)|P (100%)| S (10%)|S (100%)|
> > |----|----|----|----|----|----|----|----|----|
> > |Our MRM| 88.5 | 88.5 | 88.7|91.3|92.7|93.3|73.2|91.4|
> > |M3AE| 86.2 | 87.3 | 87.9 | 89.0 | 90.8 | 92.3 | 72.0 | 90.4|
> >
> > For fairness, we fine-tuned ViT-16 from scratch on MIMIC-CXR using M3AE. We use the default parameter setting presented in M3AE (Geng et al., 2022). From the above table, we see that M3AE generally performs worse than our MRM in all labeling ratios, especially with minimal data. The underperformance of M3AE may be attributed to the fact that it requires a large amount of multi-modal data to learn transferable joint representations for images and texts.
> >
> > ---------------------------------------
> > **Comment-5**:
> >
> > > What is the labeling ratio defined as?
> >
> > **Response-5**:
> >
> > The labeling ratio X% means that X% of the training set from a fully annotated downstream dataset are used for supervised fine-tuning. We have added this definition to the updated manuscript.
> >
> > ---------------------------------------
> > **Comment-6**:
> > > It is unclear why the weight (lambda) of the report and the image loss is equal, have the authors experimented with this weight?
> >
> > **Response-6**:
> >
> > Thanks for your kind comments. We have performed an ablation study of the controlling factor $\lambda$ in Eq. 3. We present these results in Table 5(c) in the revised manuscript. We provide an excerpt here for your consideration (all experiments are performed on NIH ChestX-ray):
> >
> > |$\lambda$|1%|10%|100%|
> > |----|----|----|----|
> > |1|79.4|84.0|85.9|
> > |2|78.6|83.5|85.4|
> > |0.5|79.0|83.7|85.9|
> >
> > The above results show that $\lambda=1$ is an optimal choice. In contrast, a smaller $\lambda$ value (i.e., 0.5) performs better than a larger value (i.e., 2.0), indicating that the MLM objective may play a more critical role than the MIM objective during the pre-training stage.
> >
> > ---------------------------------------
> >
> > Finally, we sincerely appreciate your insightful and constructive comments, and thank you again for your time and efforts in reviewing our paper. We have carefully revised the manuscript based on your comments. Please let us know if you have any further questions.

---

> > > ### Comment · Reviewer_Xf1P · 2022-11-18
> > > **thank you**
> > >
> > > Thank you for your extensive response. I believe my concerns have been addressed, and I updated my rating.

---

> ### Author Response · Authors · 2022-11-18
> **Follow up**
>
> Dear Reviewer Xf1P,
>
> Thank you for your time and effort in reading our response! We hope our response has addressed your concerns. If you still feel unclear or concerned, please let us know, and we will be more than glad to clarify and discuss any further concerns. If you feel your concerns have been addressed, please kindly consider if it is possible to update your score.
>
> Thank you!
>
> Paper1629 Authors

---

### Official Review · Reviewer_UVnZ · 2022-10-29

**Confidence:** 4
**Correctness:** 4
**Technical Novelty And Significance:** 4
**Empirical Novelty And Significance:** 4
**Recommendation:** 8

**Clarity, Quality, Novelty And Reproducibility:**



I found the paper very interesting and overall well written. To the best of my knowledge, the presented methodology is novel.







**Strength And Weaknesses:**


One of the biggest issues when developing algorithms for chest xray analysis is access to labelled data, since it is expensive for radiologists to label tens of thousands of images. Most often, labels are  then generated with rule-based NLP algorithms. These labels are however very noisy, both because of the complexity of the text data to analyse, but also because in many cases radiologists might not write in the reports about diseases that are present in the image but are not clinically relevant for the patient.

The method discussed in this paper allows to consider an alternative approach: learn transferable radiography representations using image+report data, and fine-tune the networks for downstream tasks using a low amount of labels.

The experimental section is extensive, and shows increased performances compared to related methods. Importantly for practical purposes, thanks to the pretrained model, good classifiers can be built only using 10% of the labels.

All the ablation studies I had in mind while reading the MRM section were done by the authors, which show that all components of the proposed model are necessary.
It would be interesting for me to see how the model performs with different amounts of masking probabilities. In particular I am surprised that the model performs well even when removing 75% of the patches.

The structure of the experimental section is a bit confusing the way it is split now in "Baselines" vs "Results". For example, after reading about the tasks and baselines in section 4.3.1, it would make more sense for the reader to directly see their results (section 4.5.1).


_Typos_:
* Second last paragraph of introduction: base->based
* First sentence of section 3: learns -> learn

**Summary Of The Paper:**

This paper focuses on building better models for unsupervised pretraining of deep neural networks for chest xray analysis.  This is achieved combining multi-modal information, i.e. radiological reports and chest xray images, and defining an unsupervised training procedure that predicts simultaneously masked parts of the images/reports given the non-masked part (masked record modelling).



The learned representations can be used to initialize networks for downstream tasks (classification, segmentation), and allow to outperform competing methods even when using a much lower amount of labelled data.



**Summary Of The Review:**

Overall an interesting paper with practical applications. I believe it can have an impact in the medical imaging community.

---

> ### Author Response · Authors · 2022-11-16
> **Response to Reviewer UVnZ**
>
> We sincerely appreciate your positive comments regarding the novelty, experimental performance improvements, and paper writing. We have incorporated your insightful comments during revision. In the following, we post point-by-point responses, which we hope can help resolve your concerns. If you have further questions, please do not hesitate to post them.
>
> ----------------------------
> **Comment-1**:
>
> >It would be interesting for me to see how the model performs with different amounts of masking probabilities. In particular, I am surprised that the model performs well even when removing 75% of the patches.
>
> **Response-1**:
>
> Thanks for your insightful comments. We have conducted an ablation study to investigate the impact of applying different masking ratios to radiographs. The experimental results are reported in Table 5(b) in the revised manuscript, and the associated analyses lie in the last paragraph of Sec. 4.5. We provide an excerpt here for your consideration:
>
> |$\mathcal{P}_I$|1%|10%|100%|
> |----|----|----|----|
> |75%|79.4|84.0|85.9|
> |50%|78.8|84.4|86.1|
> |0%|75.4|82.0|84.4|
>
> From the above table (all experiments are performed on NIH ChestX-ray), we find that a ratio of 75\% performs the best on the minimal labeling ratio (i.e., 1\%). In contrast, a ratio of 50\% achieves slightly better results as the labeling ratio increases. In MRM, we set the default masking ratio for radiographs to 75\% because this operation leads to fewer input patches, accelerating the pre-training process and reducing the memory cost.
>
> The insight behind applying a high masking ratio (i.e., 75%) is that it exploits the heavy spatial redundancy of radiographs. By applying a high masking ratio, we reduce the redundancy and create a surrogate task, requiring the model to understand the high-level semantics holistically.
>
> ----------------------------
> **Comment-2**:
>
> >The structure of the experimental section is a bit confusing the way it is split now in "Baselines" vs "Results". For example, after reading about the tasks and baselines in section 4.3.1, it would make more sense for the reader to directly see their results (section 4.5.1).
>
> **Response-2**:
>
> We appreciate your comments to improve our paper's organization. We have placed section 4.3 (Baselines) next to section 4.4 (Results) in the updated manuscript.
>
> ----------------------------
> **Comment-3**:
>
> > Some typos.
>
> **Response-3**:
>
> We appreciate your efforts in pointing out these typos. We have fixed them in the updated manuscript.
>
> ----------------------------
> Finally, we would like to express our sincere gratitude to you for providing constructive comments. We have carefully revised our manuscript according to your advice and comments.

---

### Author Response · Authors · 2022-11-18
**Summary of Revisions**

We sincerely appreciate the reviewers' efforts in providing constructive reviews and insightful suggestions. We have incorporated their constructive feedback as revisions to our paper. We now summarize these changes as follows:

----
### Experiments

- [UVnZ,Xf1P,JZho] *Ablation studies of different masking ratios and non-masking pretext tasks*: Table 5(b) presents the experimental results of applying different masking ratios to radiographs. Table 4 displays the ablative results of three non-masking pretext tasks: 1) w/o masked report modeling; 2) w/o masked radiograph modeling; 3) w/o masked report & radiograph modeling.

- [Xf1P,JZho] *Ablation studies of $\lambda$ in Eq. 3*: Table 5(c) presents the ablative results of the hyper-parameter $\lambda$ used in Eq.3.

- [Xf1P] *Comparison to non-medical multi-modal pre-training methods*: We compare our MRM against M3AE (Geng et al., 2022), a recent representative multi-modal pre-training methodology based on masked modeling. Table 1 presents the experimental results of M3AE on CheXpert, RSNA Pneumonia, and SIIM. We see that our MRM outperforms M3AE on all three datasets in different labeling ratios, sometimes surpassing it by large margins when the amount of labeled data is minimal.

- [JZho] *Alternative for multi-modal fusion*: Table 5(a) presents the results of two common multi-modal fusion strategies, where we compare global average pooling (our choice) with global maximum pooling. We see that GAP achieves slightly better results. It is a promising direction to investigate how to fuse multi-modal information in multi-modal pre-training in an optimal way. We will continue to investigate this topic in future work.

----
### Discussion

- [Xf1P,35Lo] *Relevant work aside from medical applications*: We have added more related literature in section 2 (Related Work). We also add a discussion section in Appendix A.1, where we showcase the differences between our MRM and masked vision-and-language pre-training developed for natural images & language.

----
### Writing

- [UVnZ] *Structure of the experimental section*: We have optimized this section to make it more friendly to readers.

- [UVnZ,35Lo] *Some typos*: We have fixed in the updated manuscript.

----
In individual responses, we also answered questions from each reviewer and made additional clarifications. We hope that our responses and revisions are helpful. We sincerely appreciate your time and efforts in helping us to improve our paper. Please do let us know if you have any additional questions!

---

### Decision · Program_Chairs · 2023-01-20

**Decision:**

Accept: poster

**Justification For Why Not Higher Score:**

This is a solid paper but the technical novelty is still somewhat limited.

**Justification For Why Not Lower Score:**

I do not recommend reject, given the overall solidness of the paper contributing an effective method for a useful problem, and strong experiments. The paper was also given positive reviews by all four reviewers.

**Metareview: Summary, Strengths And Weaknesses:**

All reviewers concurred that this paper is above the bar for acceptance. Reviewers found the work to be solid overall, with a reasonable method and strong experimental validation. Demonstrating the utility of a unified framework for masked image and language modeling for radiograph+report data is a useful contribution. Although some reviewers noted that the technical novelty is a bit limited, they still appreciated the strong evaluation demonstrating the effectiveness of the method. The author rebuttal further addressed reviewer concerns and added improved ablation studies. I agree with the reviewer assessments and recommend acceptance for this paper.


**Note From Pc:**

if the above contains the word "oral" or "spotlight" please see: "oral" presentation means -> notable-top-5% and "spotlight" means -> notable-top-25%. As stated in our emails, we are disassociating presentation type from AC recommendations